# Prognostic Value of Speckle Tracking Echocardiography-Derived Strain in Unmasking Risk for Arrhythmias in Children with Myocarditis

**DOI:** 10.3390/biomedicines12102369

**Published:** 2024-10-16

**Authors:** Nele Rolfs, Cynthia Huber, Bernd Opgen-Rhein, Isabell Altmann, Felix Anderheiden, Tobias Hecht, Marcus Fischer, Gesa Wiegand, Katja Reineker, Inga Voges, Daniela Kiski, Wiebke Frede, Martin Boehne, Malika Khedim, Daniel Messroghli, Karin Klingel, Eicke Schwarzkopf, Thomas Pickardt, Stephan Schubert, Fatima I. Lunze, Franziska Seidel

**Affiliations:** 1Department of Congenital Heart Disease—Pediatric Cardiology; Deutsches Herzzentrum der Charité, 13353 Berlin, Germany; 2Charité—Universitätsmedizin Berlin, Corporate Member of Freie Universität Berlin and Humboldt-Universität zu Berlin, 10117 Berlin, Germany; 3DZHK (German Centre for Cardiovascular Research), Partner Site Berlin, 10785 Berlin, Germany; 4Department of Medical Statistics, University Medical Center Göttingen, 37075 Göttingen, Germany; 5Clinic for Pediatric Cardiology, Heart Centre, University of Leipzig, 04109 Leipzig, Germany; 6Pediatric Cardiology, University Hospital Erlangen, 91054 Erlangen, Germany; 7Center for Congenital Heart Disease, Heart and Diabetes Center NRW, University Hospital of the Ruhr University Bochum, Medical Faculty OWL (University of Bielefeld), 32345 Bad Oeynhausen, Germany; 8Department of Pediatric Cardiology and Pediatric Intensive Care, Ludwig Maximilians University of Munich, 80336 Munich, Germany; 9Pediatric Cardiology, University Hospital Tübingen, 72076 Tuebingen, Germany; 10Department of Congenital Heart Disease and Pediatric Cardiology, University Heart Center Freiburg-Bad Krozingen, Faculty of Medicine, University of Freiburg, 79106 Freiburg, Germany; 11Department for Congenital Heart Disease and Pediatric Cardiology, University Hospital Schleswig-Holstein, 24105 Kiel, Germany; 12DZHK (German Centre for Cardiovascular Research), Partner Site Hamburg/Lübeck/Kiel, 24105 Kiel, Germany; 13Pediatric Cardiology, University Hospital Münster, 48149 Muenster, Germany; 14Pediatric Cardiology and Congenital Heart Defects, Center for Pediatrics, University Hospital Heidelberg, 69120 Heidelberg, Germany; 15Department of Pediatric Cardiology and Intensive Care Medicine, Hannover Medical School, 30625 Hannover, Germany; 16Pediatric Cardiology, University Hospital RWTH Aachen, 52074 Aachen, Germany; 17Department of Cardiology, Angiology and Intensive Care Medicine, Deutsches Herzzentrum der Charité, 10117 Berlin, Germany; 18Cardiopathology, Institute for Pathology, University Hospital Tuebingen, 72076 Tuebingen, Germany; 19Competence Network for Congenital Heart Defects, 13353 Berlin, Germany

**Keywords:** myocarditis, pediatrics, speckle tracking echocardiography, longitudinal strain, major adverse cardiac events, arrhythmias, prediction

## Abstract

**Background/Objectives:** Risk assessment in pediatric myocarditis is challenging, particularly when left ventricular ejection fraction (LVEF) is preserved. This study aimed to evaluate LV myocardial deformation using speckle-tracking echocardiography (STE)-derived longitudinal +strain (LS) and assessed its diagnostic and prognostic value in children with myocarditis. **Methods:** Retrospective STE-derived layer-specific LV LS analysis was performed on echocardiograms from patients within the multicenter, prospective registry for pediatric myocarditis “MYKKE”. Age- and sex-adjusted logistic regression and ROC analysis identified predictors of cardiac arrhythmias (ventricular tachycardia, ventricular fibrillation, atrioventricular blockage III°) and major adverse cardiac events (MACE: need for mechanical circulatory support (MCS), cardiac transplantation, and/or cardiac death). **Results:** Echocardiograms from 175 patients (median age 15 years, IQR 7.9–16.5 years; 70% male) across 13 centers were included. Cardiac arrhythmias occurred in 36 patients (21%), and MACE in 28 patients (16%). Impaired LV LS strongly correlated with reduced LVEF (r > 0.8). Impaired layer-specific LV LS, reduced LVEF, LV dilatation, and increased BSA-indexed LV mass, were associated with the occurrence of MACE and cardiac arrhythmias. In patients with preserved LVEF, LV LS alone predicted cardiac arrhythmias (*p* < 0.001), with optimal cutoff values of −18.0% for endocardial LV LS (sensitivity 0.69, specificity 0.94) and –17.0% for midmyocardial LV LS (sensitivity 0.81, specificity 0.75). **Conclusions:** In pediatric myocarditis, STE-derived LV LS is not only a valuable tool for assessing systolic myocardial dysfunction and predicting MACE but also identifies patients at risk for cardiac arrhythmias, even in the context of preserved LVEF.

## 1. Introduction

Myocarditis is a potentially life-threatening condition in pediatric patients, with an incidence of 1 to 2 cases per 100,000 children [1]. Since children show a great variety of symptoms, the correct diagnosis can be challenging. Infants and young children especially are at high risk for acute heart failure with impaired cardiac function, the need for mechanical circulatory support (MCS), and transplantation with increased mortality [2]. Cardiac arrhythmias and sudden cardiac death (SCD) are another dreaded manifestation, occurring even in patients with preserved systolic function [3]. Overall, cardiac arrhythmias appear in approximately one third of children with myocarditis and include atrial and ventricular tachycardia as well as high-grade atrioventricular block [4]. Endomyocardial biopsy (EMB) remains the gold standard for diagnosis. As an invasive procedure, it carries periprocedural risks and is prone to sampling errors. It is therefore not indicated in all patients, particularly in those with preserved or mildly reduced cardiac function. In recent years, and with the revision of the Lake Louise Criteria (LLC), cardiac magnetic resonance imaging (cMRI) has become an alternative tool with a high sensitivity and specificity for the diagnosis of myocarditis [5]. In addition to its diagnostic value, the presence of late gadolinium enhancement (LGE) appears to be an independent prognostic predictor of poor outcome in myocarditis [6,7]. However, the use of cMRI in pediatric patients with clinically suspected myocarditis may not always be feasible due to limited accessibility, lack of expertise, or simply the need for sedation in younger children. Novel two-dimensional speckle tracking echocardiography (STE)-derived strain identifies left ventricular (LV) regional myocardial deformation. Global longitudinal strain (GLS) not only serves as a parameter of impaired systolic myocardial function but is also more sensitive than conventional global LV ejection fraction (LVEF) when unmasking subtle LV dysfunction in adults with myocarditis [8,9]. A recent study demonstrated that LV longitudinal strain (LS) was as an independent predictor of major adverse cardiac events (MACE) in adults with myocarditis [10].

Therefore, we aimed to evaluate LV myocardial deformation using STE-derived LS in children with cMRI- and/or EMB-confirmed diagnosis of myocarditis, as part of the multicenter registry for pediatric myocarditis “MYKKE”.

We hypothesized that in pediatric patients with myocarditis, STE-derived LS

serves as an accurate diagnostic parameter for assessing myocardial systolic dysfunction;provides additional prognostic value, especially in patients with preserved LVEF.

## 2. Materials and Methods

The “MYKKE” Registry is a multicenter, prospective, long-term registry for suspected pediatric myocarditis that provides a research platform for clinical trials investigating the epidemiology, clinical entity, diagnostic and therapeutic approaches, and outcomes of pediatric myocarditis. The registry is hosted and technically managed by the Competence Network for Congenital Heart Defects. Ethical approval was first obtained at the initiating center (Deutsches Herzzentrum Berlin, Berlin, Germany) by the Ethics Committee of Charité—Universitätsmedizin Berlin (EA2/074/13) and subsequently confirmed by the local authorities of all collaborating centers (ClinicalTrials.gov Identifier: NCT02590341) according to the Declaration of Helsinki. Since 2013, 30 medical centers in Germany, Switzerland, and Austria have become participating partners. Within the “MYKKE” registry, we designed a sub-study performing retrospective echocardiographic strain analysis at initial admission in children with confirmed myocarditis to evaluate its diagnostic and prognostic value.

### 2.1. Inclusion Criteria

All patients met the general “MYKKE” inclusion criteria published previously [4]. For this specific sub-study, only patients with the following inclusion criteria were enrolled:(a)Diagnosis of myocarditis confirmed by EMB and/or cMRI;(b)Transthoracic echocardiography with apical 4-chamber view (CV) of sufficient quality to assess STE-derived LV LS.

We excluded patients with myocarditis after SARS-CoV-2-vaccination, pre-existing complex congenital heart disease, or arrhythmias.

### 2.2. Data Monitoring

Clinical data were obtained from the online study database as previously described [4]. Local protocols were used for cMRI analysis and image interpretation was performed by the respective centers. If endomyocardial biopsy was indicated, samples were analyzed by a single DAkkS accredited laboratory (Cardiopathology, Institute of Pathology and Neuropathology, University Hospital Tuebingen, Tuebingen, Germany). The diagnosis of myocarditis was confirmed according to the established histopathologic and immunohistochemical criteria [11,12]. Primary outcomes were major adverse cardiac events (MACE), defined as a composite outcome of the need for mechanical circulatory support (MCS), cardiac transplantation, and/or cardiac death. Of particular interest was the occurrence of cardiac arrhythmias during hospitalization or after discharge, including sustained and non-sustained ventricular tachycardia (sVT; nsVT), ventricular fibrillation (VF), and atrioventricular blockage (AVB) III°. The occurrence of cardiac arrhythmias was monitored both during the initial hospitalization and after discharge, either at re-hospitalization or outpatient visits. Cardiac arrhythmias were documented using Holter ECG and/or continuous cardiorespiratory monitoring.

### 2.3. Conventional and Speckle Tracking Echocardiography

Participating “MYKKE” sites provided pseudonymized echocardiograms from the patients’ first admission in DICOM format, which were stored on a central research server. For each patient, the echocardiographic study closest in time to symptom onset was selected for analysis. The senior cardiologist (F.I.L.) trained in cardiac imaging prepared the study protocol with the echocardiographic indices. All conventional and STE-derived strain indices were measured retrospectively from the DICOM recordings by two cardiologists (F.I.L. and N.R.) blinded to the clinical data using a commercially available offline software package (Tomtec Imaging Systems Inc., Unterschleissheim, Germany). Conventional echocardiographic LV size and function assessment included LV diameters, fractional shortening (FS), LVEF, wall thicknesses, and LV mass (LVM) based on the Devereux formula and were assessed using the M-mode either at parasternal long axis or short axis. Z-scores were calculated indexed to the patient’s body surface area (BSA) according to Kampmann et al. [13]. Global LV systolic function was quantified by LVEF and FS based on the Teichholz method [14]. According to the European Society of Cardiology’s definition of heart failure, the severity of the LV systolic dysfunction was defined as follows:Preserved LVEF ≥ 50% (pEF)Mildy reduced LVEF 41–49% (mrEF)Reduced EF ≤ 40% (rEF) [15].

Regional STE-derived LS indices were obtained in the apical 4-CV for six LV segments (basal inferior/septal, mid inferior/septal, apical septal, basal anterior/lateral, mid anterior/lateral, and apical lateral) at the endocardial, midmyocardial, and epicardial levels. LV peak LS was calculated as the average of the six segments for each cardial layer. For STE-derived strain analysis, only 4-CV images of good quality with at least 4 measurable segments in the endocardial, midmyocardial, and epicardial layers were included. After placing tracing points at the endocardial and epicardial borders at systole and end-diastole, the automated border detection was visually inspected frame by frame across the cardiac cycles. Manual adjustments to the tracing points were made as necessary for optimal tracking. LV dyssynchrony was quantified using the maximum opposing wall delay (maxOWD), which was automatically calculated by the TomTec Arena software (Tomtec TTA2.51.4) as the difference in time to maximum LS peak (TTPmax) of the septal and lateral walls.

### 2.4. Statistical Analysis

Descriptive data are presented as medians with interquartile range (IQR) for continuous variables and as counts with percentages for categorical variables. Kruskal–Wallis rank sum test or Pearson’s Chi-squared and Fisher exact tests were used to compare continuous and categorical variables in these groups, respectively. Age- and sex-adjusted multivariable logistic regression models were used to evaluate echocardiographic indices as predictors for the primary outcomes MACE and cardiac arrhythmias. Odds ratio (OR) and their 95% confidence intervals (CI) are reported. Receiver operating characteristic (ROC) analyses were performed to determine potential cutoff values for endocardial and midmyocardial peak LS associated with cardiac arrhythmias in pediatric myocarditis patients. Youden’s J statistic was applied to identify the optimal thresholds. Using these cutoff values, Kaplan–Meier survival curves were plotted to illustrate the duration of freedom from cardiac arrhythmias. A *p*-value < 0.05 was considered statistically significant. All calculations with LS indices were based on absolute numbers. Data were analyzed using R version 4.3.1.

## 3. Results

### 3.1. Clinical Characteristics

Echocardiograms from 175 patients recruited from 13 centers were considered eligible (see Figure 1). Table 1 shows the clinical characteristics of the final study cohort stratified by M-mode-derived LVEF. Overall, the median age at initial presentation was 15 (7.9-16.5) years; 123 patients (70%) were male. Patients with rEF were significantly younger and evenly distributed between the sexes, whereas patients with pEF were predominantly adolescents and male (*p* < 0.001). Chest pain was the most prevalent symptom in patients with pEF. As expected, dyspnea and symptoms of severe heart failure were more common in patients with rEF (*p* < 0.001). Accordingly, the brain natriuretic peptide (BNP) or its N-terminal prohormone (NT-proBNP) were more frequently elevated in these patients (*p* < 0.001). No difference in troponin elevation was observed between the groups. Cardiac MRI was performed in 75% (n = 132) of the total study cohort. Of these, 88% (n = 116) had findings suggestive of myocarditis. In the remaining cases, where cMRI did not confirm myocarditis according to the criteria, the EMB verified the diagnosis. Overall, EMB was obtained in 125 patients (71%). The diagnosis of myocarditis was confirmed according to the established histopathologic/immunohistochemical criteria in 122 patients (98%): acute lymphocytic myocarditis in 33 patients (26%), chronic/healing lymphocytic myocarditis in 73 (58%), and healed myocarditis in 16 (13%) patients. Patients with rEF had a significantly higher incidence of the previously defined MACE including MCS, heart transplantation, and mortality (*p* < 0.001). In the overall cohort, 28 patients (16%) had at least one MACE at a median of 7 (IQR 2–97) days after admission. Thirty-six patients (21%) experienced cardiac arrhythmias at a median of 2 (IQR 2–16) days after admission. Importantly, this was independent of systolic cardiac function as determined by LVEF (*p* = 0.200). The most frequently monitored arrhythmias were VTs in a total of 30 patients (17%; sVT n = 15/nsVT n = 15), followed by VF in 5 (3%) and AVBIII° in 2 (1%) patients. Only one patient had more than one type of arrhythmias (VT and VF). Clinical follow-up data were available for 147 patients from the study cohort, obtained either through rehospitalization or outpatient visits. Post-discharge cardiac arrhythmias were documented in four patients using Holter ECG and/or cardiorespiratory monitoring.

### 3.2. Conventional and STE-Derived Strain Analyses

Echocardiography was performed at a median of 0 (IQR 0–1) days after admission. Table 2 shows the conventional echocardiographic assessment of M-mode-derived indices and LV function. Patients with rEF were more likely to have LV dilatation (*p* < 0.001) and increased BSA-indexed LVM (*p* < 0.001). Peak LS of all cardial layers differed significantly among the three LVEF-stratified groups, with strain measures being most impaired in patients with rEF (all *p* < 0.001) (see Table 3). The same was true for the endo- and midmyocardial maxOWD: patients with rEF had the most pronounced wall dyssynchrony (*p* = 0.025 and *p* = 0.024, respectively). Figure 2 illustrates the high correlation between echocardiographic strain indices and LVEF (correlation coefficient r > 0.8). Peak LS also correlated with BSA-indexed LVM, and Z-score of LV end-diastolic diameter (LVEDd) (both r > 0.5).

### 3.3. Predictive Value of STE-Derived Strain

Patients who experienced MACE had significantly lower LV LS in all cardial layers compared with those without (endocardial LS −8% vs. −20%; midmyocardial LS −6.8% vs. −16.3%; epicardial LS −5.2% vs. −13.6%, all *p* < 0.001). In an age- and sex-adjusted multivariable logistic regression model, reduced layer-specific LV LS, as well as an impaired LVEF, an increase in Z-score of LVEDd, and BSA-indexed LVM, were associated with increased risk of MACE (see Table 4). Given that MACE were rarely reported in patients with pEF and mrEF, we did not perform separate analyses for each of the LVEF-stratified groups. By analogy, all layer-specific LV LS indices were significantly worse in patients with cardiac arrhythmias compared with those without (endocardial LS −13% vs. −21%; midmyocardial LS −9.8% vs. −16.3%; epicardial LS −8.1% vs. −13.2%, all *p* < 0.001). Using the adjusted multivariable logistic regression model, reduced layer-specific LV LS indices were associated with an elevated risk of cardiac arrhythmias (all *p* < 0.001) (see Table 5). Exemplarily, the odds of cardiac arrhythmias decreased by 22% for every 1% increase in midmyocardial LV LS in the overall cohort (95% CI, 0.7–0.86; *p* < 0.001). Figure 3 illustrates the predicted probability for the occurrence of MACE (Figure 3A) and cardiac arrhythmias (Figure 3B) as a function of LV LS and based on the adjusted regression model. An endocardial LV LS of −10.6% corresponds to a 50% probability of cardiac arrhythmias in a male at the age of 14.9 years. The same is true for a female patient with an endocardial peak LS of −13.3%. In the overall cohort, arrhythmias were further associated with decreased LVEF, as well as LV dilatation and increased BSA-indexed LVM (*p* = 0.008, *p* = 0.008, *p* < 0.001; respectively). Separate analyses were performed for each LVEF-stratified group, while the adjusted regression model in the mrEF group was not statistically valid with only three arrhythmic events and therefore is not shown. While the conventional echocardiographic parameters LVEF, LVEDd, and BSA-indexed LVM showed a significant association with cardiac arrhythmias in the overall cohort, this was not the case when patients with pEF and rEF were considered separately (see Table 5). Of note, in patients with pEF, the impairment of all layer-specific LV LS indices was significantly associated with the occurrence of cardiac arrhythmias (all *p* < 0.001). As an example, Figure 4 illustrates the differences in endocardial LS between two male patients with and without the occurrence of ventricular tachycardia, both with normal LVEF. In the ROC curve analyses with the entire cohort, a threshold of endocardial LV LS at −18.3% (corresponding to a sensitivity of 0.86 and a specificity of 0.58) and of midmyocardial LV LS at −13.3% (corresponding to a sensitivity of 0.75 and a specificity of 0.63) showed adequate performance for predicting arrhythmias. In patients with pEF only, the ROC curve analyses found the optimal cutoff values for detecting patients at greater risk for cardiac arrhythmias for endo- and midmyocardial LV LS to be –18.0% (corresponding to a sensitivity of 0.69 and a specificity of 0.94) and –17.0% (corresponding to a sensitivity of 0.81 and a specificity of 0.75), respectively. Kaplan–Meier analysis further illustrated that most cardiac arrhythmias occurred within the first few weeks of hospitalization, with only a small number of events observed after six months (see Figure 5).

### 3.4. Reproducibility

Inter- and intra-observer agreement were assessed using the standard Bland–Altman method, including 10 studies each in a blinded manner. Mean endo-, midmyo-, and epicardial peak LS showed adequate inter-observer (F.I.L; N.R.) variability with absolute mean differences of 0.9% (SD ± 0.8), 0.5 (SD ± 0.6), and 0.1% (SD ± 1.2), respectively.

## 4. Discussion

In this retrospective multicenter study, we evaluated LV myocardial deformation using STE-derived LS in 175 pediatric patients with cardiac MRI- or EMB-confirmed myocarditis. LV LS showed a high correlation with conventional echocardiographic parameters of cardiac systolic function, and impaired LV LS was associated with the occurrence of MACE and cardiac arrhythmias.

### 4.1. STE-Derived Strain and MACE

MACE, including heart failure requiring MCS, heart transplantation, and mortality, occurred in 16% of patients. These were predominantly young patients with severely impaired systolic function on echocardiogram as determined by reduced LVEF and increased LVEDd. Impaired systolic LV function and young age have previously been defined as risk factors for a severe course of the disease [2]. In our study, reduced LV LS was an independent predictor of MACE, underscoring the utility of strain imaging in the assessment of myocardial function. This is supported by some recently published studies in adults [10,16,17]. It has further been shown that adult patients with acute myocarditis and preserved LVEF exhibit impaired LS compared with healthy controls [18]. In these patients, reduced LS was further associated with the occurrence of MACE, including cardiac death, life-threatening arrhythmias, and heart failure [19]. Our data did not show a significant association between MACE and LV LS when considering only patients with pEF. This was probably due to the fact that cardiac arrhythmias were not included in the composite outcome of MACE but were considered separately, which resulted in only one event of MACE in our cohort of patients with pEF.

### 4.2. STE-Derived Strain and Arrhythmias

Consistent with the published data on pediatric myocarditis, we found a 21% likelihood of cardiac arrhythmias in the overall cohort [20]. Notably, this did not differ significantly between the LVEF-stratified subgroups. Thus, arrhythmias occurred regardless of the systolic function as assessed by LVEF, as previously reported [21]. Cardiac arrhythmias are a significant burden in myocarditis and can manifest as SCD [22]. The occurrence of arrhythmias in myocarditis patients is associated with poorer in-hospital outcomes, increased lengths of stay, and high costs [23]. Therefore, reliable diagnostic and prognostic tools are crucial to detect patients at risk. Our study showed that LV LS not only correlated well with conventional echocardiographic parameters LVEF and LVEDd, but was also useful in predicting cardiac arrhythmias, especially in the setting of preserved LVEF. Since up to one third of patients with myocarditis present with preserved LVEF with an overall good prognosis, the occurrence of cardiac arrhythmias or progression to dilative cardiomyopathy in some cases makes it important to assess subclinical myocardial dysfunction at an early stage [24,25,26]. The precise mechanism behind arrhythmias in acute myocarditis remains unclear. Potential mechanisms include myocardial cell death leading to fibrosis, immune-mediated cell damage, and therefore inflammation and edema, as well as the proarrhythmic effects of cytokines [27]. In a retrospective, single-center study, LS showed adequate diagnostic performance in detecting immuno-histologically confirmed myocardial inflammation in adult patients with preserved LVEF on cardiac MRI [28]. This finding was further supported in a pediatric cohort with cardiac MRI-confirmed myocarditis and preserved LVEF, aligning with previously reported LS reference values [29,30]. LS was regionally impaired in areas of edema and residual fibrosis identified by cardiac MRI, consistent with the presence of focal myocardial dysfunction in patients with preserved LVEF. This may be supported by the observed correlation of impaired strain indices with increased BSA-indexed LVM as a marker of myocardial edema. For that matter, partial myocardial involvement seems to be sufficient to cause arrhythmias without compromising the global systolic function. This could explain why, in our study, LS was a strong predictor of cardiac arrhythmias in patients with preserved LVEF, given the regional assessment of myocardial deformation. The superiority of STE-derived strain in detecting the subtle and regional impairment of cardiac function was also described in pediatric patients with multisystem inflammatory syndrome in children associated with COVID-19 (MIS-C) and in those with subclinical cardiomyopathies such as hypertrophic and dilated cardiomyopathies (DCM) [31,32]. In fifty children with recovered primary DCM, LV layer-specific myocardial strain was significantly reduced compared with a control cohort of healthy children despite similar conventional echocardiographic parameters [33].

Pruitt et al. found in their single-center, retrospective cohort study a cutoff value for global LS of worse than −11% to be a predictor of arrhythmias in pediatric patients hospitalized for myocarditis, regardless of LVEF [20]. In most cases, the diagnosis of myocarditis was based on clinical evaluation, with only a few cases confirmed by cMRI and/or EMB. In our study, the respective cutoff values for the entire cohort were higher with endocardial LV LS at −18.3% and midmyocardial LV LS at −13.3%. This may be explained by the fact that our study cohort was predominately composed of patients with pEF, resulting in an overall better mean LVEF of 52%. Another explanation may lie in methodological differences, as our LS assessment considered all three cardial layers separately. However, when comparing patients with (n = 23) and without (n = 42) arrhythmias, Pruitt et al. did not observe a statistically significant difference in LVEF. In contrast to their findings, we found LVEF to be a significant predictor of arrhythmias in the adjusted regression model when considering the overall cohort, which may be due to the larger cohort size in our study.

In our study, endocardial and epicardial maxOWD as a marker of LV dyssynchrony had prognostic value in predicting cardiac arrhythmias in the overall cohort. It has previously been shown that inhomogeneous mechanical contraction is related to an increased risk of ventricular arrhythmias in patients with dilated cardiomyopathy [34]. In adult patients with acute myocarditis and pEF, STE-derived mechanical dispersion was significantly prolonged in those with supraventricular tachycardia compared with those without [35]. Future studies should focus on the characterization of this novel STE-derived parameter in the setting of inflammatory myocardial disease to assess its diagnostic and prognostic utility as well as pediatric reference values.

### 4.3. STE-Derived Strain as a Tool for Clinical Decision Making

Significantly, our study is the first to demonstrate that in pediatric patients with cMRI- and/or EMB-confirmed myocarditis and preserved LVEF, LV LS serves as a strong predictor of cardiac arrhythmias, highlighting its potential prognostic value. Our data indicate that patients with preserved LVEF but midmyocardial LV LS worse than −17.0% at the initial echocardiography upon admission require closer surveillance for arrhythmias, both during and after hospitalization. Conventional 2D echocardiography is part of the routine diagnostic workflow in pediatric patients with cardiac symptoms and offers advantages such as widespread availability, low cost, the ability for multiple re-evaluations, and virtually no side effects. Incorporating STE-derived LS into the standard echocardiographic protocol may improve its diagnostic accuracy and identify patients at risk, especially in the setting of preserved LVEF. Future studies should prospectively validate the identified cut-off values in real-time clinical settings. Our results may also encourage future research on the predictive value of STE-derived strain in other pediatric cohorts with preserved systolic function, such as those in the early or recovered stages of cardiomyopathies. Analyzing STE-derived strain in follow-up echocardiograms of patients with apparent recovery from myocarditis, as assessed by conventional echocardiography, may identify those at long-term risk and should be explored further.

### 4.4. Limitations

The retrospective nature of the echocardiographic analyses and potential variability in imaging quality across centers must be acknowledged as limitations. Due to the retrospective nature of the study, the echocardiograms analyzed did not follow a study protocol and therefore often lacked sufficient LV 2- and 3-CV images. As a result, LV LS could only be assessed in the LV 4-CV, potentially resulting in the omission of other affected LV segments. Variations in the quality of echocardiography and protocols among participating centers also limited the assessment of diastolic function. Investigating the relationship between LV diastolic dysfunction and LS may offer further insights and should be considered in future research. In general, the limitations of strain echocardiographic imaging are intervendor variability, its dependence on frame rate, and the potential impact of extrinsic mechanical factors on the intra- and inter-observer reproducibility of STE parameters [36,37]. Furthermore, we did not consider detailed clinical follow-up data, which could provide additional insights into the prognostic value of LS in pediatric myocarditis patients.

## 5. Conclusions

In this retrospective multicenter study involving 175 pediatric patients with myocarditis confirmed by cardiac MRI or EMB, STE-derived LS emerged as a diagnostic tool and robust predictor of both MACE and cardiac arrhythmias, underscoring its utility beyond conventional echocardiographic parameters. Even in patients with preserved LVEF, impaired midmyocardial LS worse than −17.0% identified those at increased risk of cardiac arrhythmias, emphasizing the importance of close monitoring in this population. These findings highlight the potential of strain imaging to improve risk assessment and management strategies for pediatric myocarditis patients.

## Figures and Tables

**Figure 1 biomedicines-12-02369-f001:**
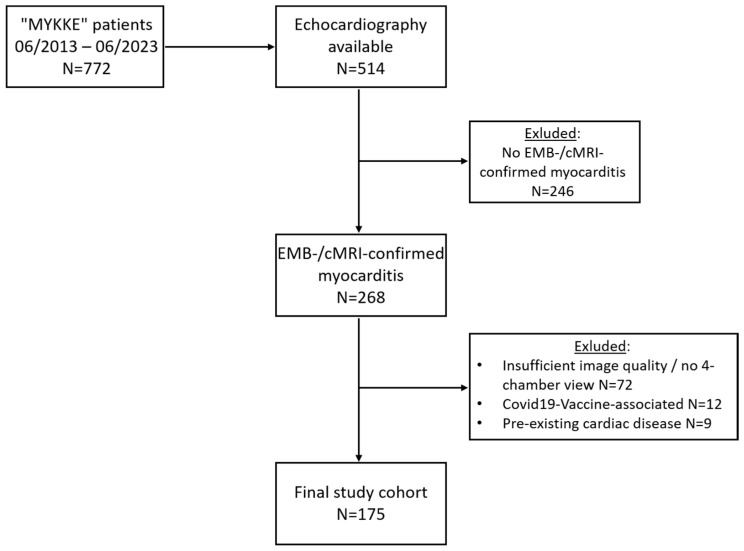
**Study flow chart.** cMRI—cardiac magnetic resonance imaging; EMB—endomyocardial biopsy.

**Figure 2 biomedicines-12-02369-f002:**
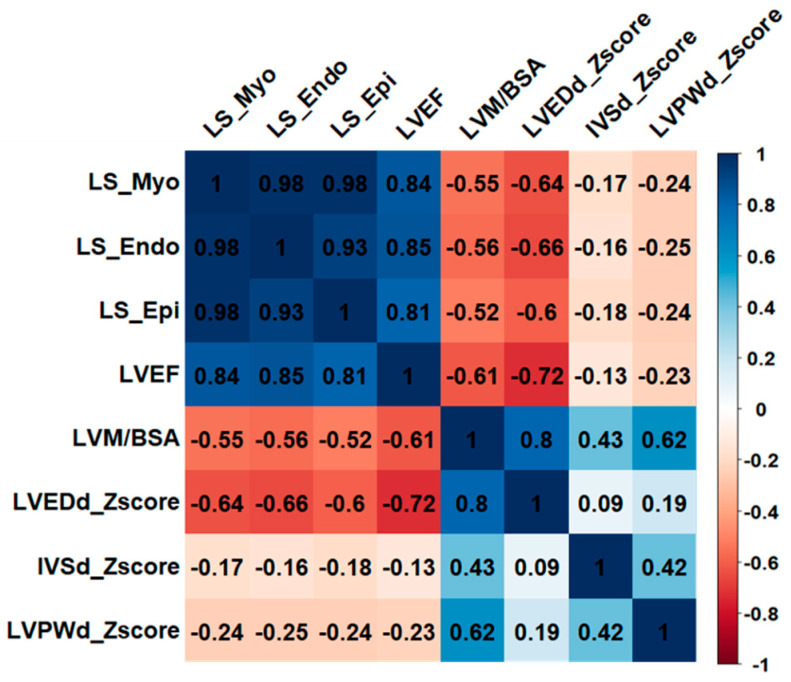
Correlation plot with a selected set of echocardiographic indices. The correlation coefficient r is presented. Values above 0.5 were considered to indicate a moderate correlation, while values greater than 0.7 were regarded as indicating a high correlation. Strain measures were based on absolute numbers. BSA—body surface area; IVSd—interventricular septal thickness at end-diastole; LS—longitudinal strain; LVEDd—left ventricular end-diastolic diameter; LVEF—left ventricular ejection fraction; LVM—left ventricular mass; LVPWd—left ventricular posterior wall thickness at end-diastole.

**Figure 3 biomedicines-12-02369-f003:**
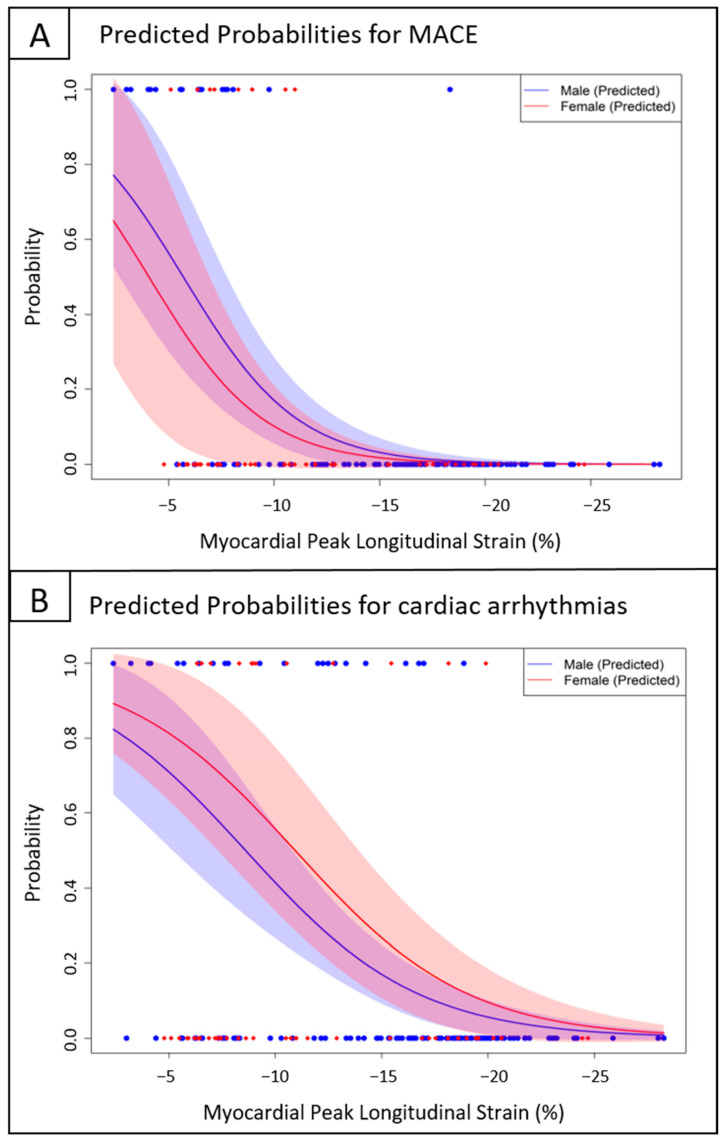
Predicted probabilities of major adverse cardiac events (MACE) (**A**) and cardiac arrhythmias (**B**) are shown as a function of mean myocardial peak longitudinal strain by sex for the median cohort age (14.9 years), based on the adjusted regression models (see Table 4 and Table 5).

**Figure 4 biomedicines-12-02369-f004:**
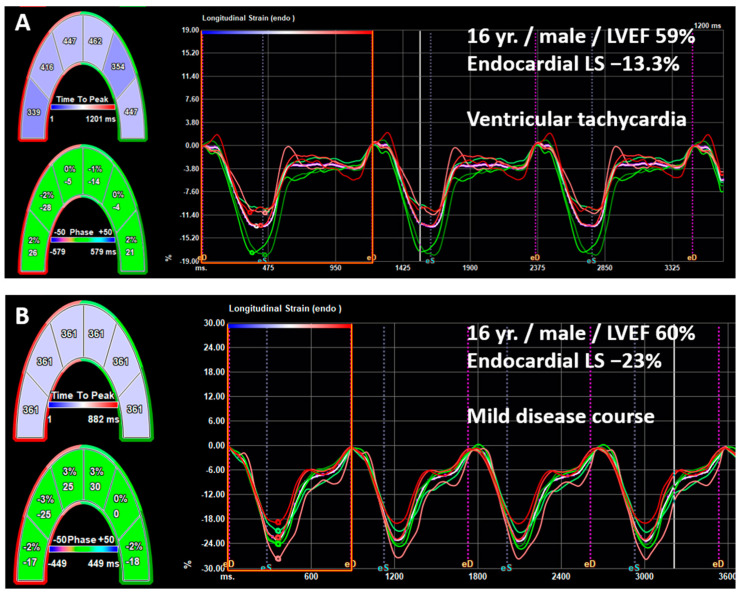
Exemplary measurements of endocardial longitudinal strain (LS) in two 16-year-old male patients, both with normal LVEF. The patient experiencing ventricular tachycardia (**A**) shows a significant decrease in endocardial LS compared with the patient without the occurrence of cardiac arrhythmias (**B**).

**Figure 5 biomedicines-12-02369-f005:**
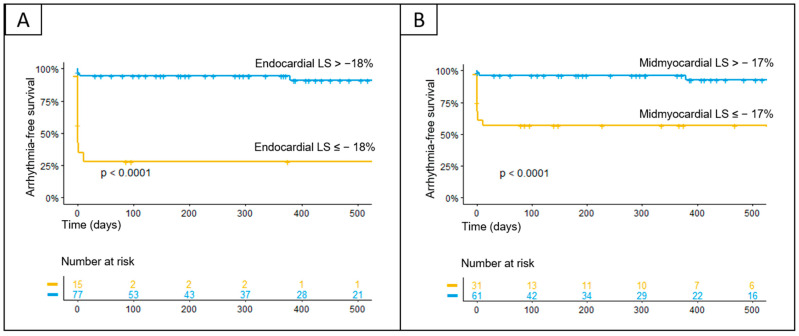
Kaplan–Meier analyses for the occurrence of cardiac arrhythmias according to the endocardial left ventricular longitudinal strain (LV LS) (**A**) and midmyocardial LV LS (**B**). The cutoff values were determined based on regression models and calculated using receiver operating characteristic (ROC) analyses and Youden’s J index.

**Table 1 biomedicines-12-02369-t001:** Clinical characteristics at initial admission stratified by echocardiographic LVEF.

Characteristics	Overall, N = 175	pEF (LVEF ≥ 50%), N = 93	mrEF (LVEF 41–49%), N = 23	rEF (LVEF ≤ 40%), N = 59	*p*-Value
**Age (years)**	15.0 (7.9, 16.5)	16.1 (14.6, 17.0)	14.6 (9.6, 16.7)	2.8 (1.2, 12.9)	**<0.001**
**Height (cm)**	168 (126, 178)	175 (167, 181)	167 (136, 176)	95 (78, 156)	**<0.001**
(Missing)	2	1	0	1	
**Weight (kg)**	59 (24, 75)	67 (57, 78)	65 (35, 77)	14 (9, 46)	**<0.001**
**Sex, n (%)**					**<0.001**
Male	123 (70%)	77 (83%)	15 (65%)	31 (53%)	
Female	52 (30%)	16 (17%)	8 (35%)	28 (47%)	
**Symptoms, n (%)**					
Fever	79 (45%)	47 (51%)	8 (35%)	24 (41%)	0.3
Shortness of breath	83 (47%)	21 (23%)	13 (57%)	49 (83%)	**<0.001**
Angina Pectoris	94 (54%)	76 (82%)	12 (52%)	6 (10%)	**<0.001**
Fatigue	125 (71%)	53 (57%)	16 (70%)	56 (95%)	**<0.001**
NYHA classification					**<0.001**
NYHA I	92 (53%)	78 (84%)	9 (39%)	5 (8.5%)	
NYHA II	24 (14%)	12 (13%)	9 (39%)	3 (5.1%)	
NYHA III	19 (11%)	2 (2.2%)	2 (8.7%)	15 (25%)	
NYHA IV	35 (20%)	1 (1.1%)	3 (13%)	31 (53%)	
n.a.	5 (2.9%)	0 (0%)	0 (0%)	5 (8.5%)	
**Electrocardiogram, n (%)**					
ST-elevation	65 (37%)	44 (47%)	11 (48%)	10 (17%)	**<0.001**
(Missing)	1	0	0	1	
ST-depression	24 (14%)	11 (12%)	5 (22%)	8 (14%)	0.5
(Missing)	1	0	0	1	
T-wave inversion	67 (39%)	27 (29%)	13 (57%)	27 (47%)	**0.019**
(Missing)	2	1	0	1	
**Laboratory Analysis, n (%)**					
Leukocytosis	44 (25%)	19 (20%)	5 (22%)	20 (34%)	0.4
(Missing)	1	1	0	0	
Elevated C-reactive protein	107 (61%)	71 (76%)	14 (61%)	22 (37%)	**<0.001**
(Missing)	3	2	0	1	
Elevated BNP/NT-proBNP	128 (73%)	57 (61%)	15 (65%)	56 (95%)	**<0.001**
(Missing)	15	11	3	1	
Elevated troponins	151 (86%)	81 (87%)	20 (87%)	50 (85%)	0.5
(Missing)	2	1	1	0	
**Cardiac MRI, n (%)**	N = 132	N = 86	N = 16	N = 30	
LGE	102 (77%)	72 (84%)	12 (75%)	18 (60%)	**0.012**
Edema	74 (58%)	48 (57%)	12 (80%)	14 (50%)	0.052
(Missing)	5	2	1	2	
MRI confirmed myocarditis	105 (80%)	73 (85%)	14 (88%)	18 (60%)	0.064
**EMB, n (%)**	N = 125	N = 47	N = 21	N = 57	
EMB confirmed myocarditis	122 (98%)	45 (96%)	20 (95%)	57 (100%)	0.100
Virus detection	51 (41%)	10 (21%)	8 (38%)	33 (58%)	**<0.001**
**Complications, n (%)**					
MACE overall	28 (16%)	1 (1.1%)	2 (9%)	25 (42%)	**<0.001**
MCS	27 (15%)	0 (0%)	2 (9%)	25 (42%)	**<0.001**
Heart transplantation	11 (6%)	0	0	11 (19%)	**<0.001**
Mortality	6 (3%)	1 (1%)	2 (9%)	3 (5%)	0.110
**Cardiac arrhythmias, n (%)**	36 (21%)	16 (17%)	3 (13%)	17 (29%)	0.2
Ventricular tachycardia	30 (17%) *	14 (14%) *	2 (9%)	14 (24%)	
Ventricular fibrillation	5 (3%) *	2 (2%) *	1 (4%)	2 (3%)	
AVB III°	2 (1%)	1 (1%)	0 (0%)	1 (2%)	

Data are reported as medians (interquartile range) or counts with percentages; percentages are rounded. The denominator of percentages is the total sample size, unless specified. The overall cohort was further categorized by M-mode-derived LVEF into the following groups: patients with preserved LVEF (pEF; LVEF ≥ 50%), mildly reduced LVEF (mrEF; LVEF 41–49%), and reduced LVEF (rEF; LVEF ≤ 40%). Depending on the variable, Kruskal–Wallis rank sum test or Pearson’s Chi-squared test/Fisher’s exact test were used for statistical comparison. *p*-value of <0.05 was considered statistically significant. BNP—brain natriuretic peptide; EMB—endomyocardial biopsy; LGE—late gadolinium enhancement; MRI—magnetic resonance imaging; n.a—not applicable; NT-proBNP—N-terminal prohormone of brain natriuretic peptide; NYHA—New York Heart Association. * One patient experienced ventricular tachycardia and fibrillation.

**Table 2 biomedicines-12-02369-t002:** Conventional echocardiographic indices stratified by LVEF.

Indices	Overall, N = 175	pEF (LVEF ≥ 50%), N = 93	mrEF (LVEF 41–49%), N = 23	rEF (LVEF ≤ 40%), N = 59	*p*-Value
Z-score LVEDd	1.3 (−0.2–4.2)	0.3 (−0.6–1.1)	0.8 (−0.4–2.5)	6.1 (3.8–7.9)	**<0.001**
Z-score IVSd	1.6 (0.6–2.5)	1.4 (0.5–2.4)	2.2 (1–3.3)	1.6 (1–2.6)	**0.033**
Z-score LVPWd	1.3 (0.6–2.6)	1.1 (0.6–1.9)	1.9 (0.8–2.9)	1.6 (0.5–2.9)	**0.018**
LVEF (%)	52 (30–59)	59 (55–64)	44 (42–48)	26 (19–30)	**<0.001**
FS (%)	27 (14–32)	31 (29–35)	22 (21–24)	12 (8–14)	**<0.001**
LVM/BSA	103 (80–135)	88 (76–103)	115 (88–124)	153 (120–178)	**<0.001**

Data are reported as medians (interquartile range). Z-scores were calculated indexed to the patient’s body surface area (BSA) according to Kampmann et al. [13]. The overall cohort was further categorized by M-mode derived LVEF into the following groups: patients with preserved LVEF (pEF; LVEF ≥ 50%), mildly reduced LVEF (mrEF; LVEF 41–49%) and reduced LVEF (rEF; LVEF ≤ 40%). Kruskal-Wallis rank sum test was performed for statistical comparison. *p*-value of < 0.05 was considered statistically significant. BSA—body surface area; FS—fractional shortening; IVSd—interventricular septal thickness at end-diastole; LVEDd—left ventricular end-diastolic diameter; LVEF—LV ejection fraction; LVM—LV mass; LVPWd—LV posterior wall thickness at end-diastole.

**Table 3 biomedicines-12-02369-t003:** STE-derived longitudinal strain indices stratified by LVEF.

Indices	Overall, N = 175	pEF (LVEF ≥ 50%), N = 93	mrEF (LVEF 41–49%), N = 23	rEF (LVEF ≤ 40%), N = 59	*p*-Value
Endocardial LV LS (%)	−18 (−10, −23)	−23 (−20, −24)	−15 (−13, −20)	−9 (−7, −11)	**<0.001**
Endocardial maxOWD (ms)	61 (39, 94)	58 (39, 81)	45 (31, 78)	81 (40, 129)	**0.025**
Midmyocardial LV LS (%)	−15.3 (−7.9, −19)	−18.6 (−16.3, −20.1)	−12.4 (−10.5, −15.6)	−7.2 (−5.9, −8.2)	**<0.001**
Midmyocardial maxOWD (ms)	62 (37, 95)	58 (36, 80)	66 (42, 92)	80 (41, 125)	**0.024**
Epicardial LV LS (%)	−11.6 (−6.8, −15.7)	−15.4 (−13.7, −17.3)	−9.2 (−7.9, −12.9)	−5.7 (−4.7, −6.9)	**<0.001**
Missing (n)	1	1	0	0	
Epicardial maxOWD (ms)	81 (53, 124)	74 (53, 102)	84 (60, 107)	102 (46, 161)	0.074
Missing (n)	2	1	1	0	

Data are reported as medians (interquartile range). The overall cohort was further categorized by M-mode derived LVEF into the following groups: patients with preserved LVEF (pEF; LVEF ≥ 50%), mildly reduced LVEF (mrEF; LVEF 41–49%) and reduced LVEF (rEF; LVEF ≤ 40%). Kruskal-Wallis rank sum test was performed for statistical comparison. *p*-value of < 0.05 was considered statistically significant. LS—longitudinal strain; maxOWD—maximum opposing wall delay.

**Table 4 biomedicines-12-02369-t004:** Age- and sex-adjusted multivariable logistic regressions by echocardiographic indices for major adverse cardiac events (MACE) in the overall cohort.

Indices	OR	95% CI	*p*-Value
Endocardial LV LS	0.74	0.63, 0.83	**<0.001**
Endocardial maxOWD	1.00	1.00, 1.01	0.10
Midmyocardial LV LS	0.69	0.57, 0.80	**<0.001**
Midmyocardial maxOWD	1.00	1.00, 1.01	0.2
Epicardial LV LS	0.69	0.56, 0.81	**<0.001**
Epicardial maxOWD	1.00	1.00, 1.01	**0.044**
LVEF	0.89	0.83, 0.93	**<0.001**
Z-score LVEDd	1.24	1.08, 1.45	**0.004**
LVM/BSA	1.02	1.01, 1.03	**0.006**

Absolute values of strain indices were used for statistical analysis. The composite outcome MACE included mechanical circulatory support, heart transplantation, and death. *p*-value of < 0.05 was considered statistically significant. CI—confidence interval; BSA—body surface area; LS—longitudinal strain; LVEDd—left ventricular end-diastolic diameter; LVEF—left ventricular ejection fraction; LVM—left ventricular mass; maxOWD—maximum opposing wall delay, OR—odds ratio.

**Table 5 biomedicines-12-02369-t005:** Age- and sex-adjusted multivariable logistic regressions by echocardiographic indices for cardiac arrhythmias.

Indices	Overall Cohort, N = 175			pEF (LVEF ≥ 50%), N = 93			rEF (LVEF ≤ 40%), N = 59		
	OR	95% CI	*p*-Value	OR	95% CI	*p*-Value	OR	95% CI	*p*-Value
Endocardial LV LS	0.81	0.74, 0.88	**<0.001**	0.60	0.46, 0.74	**<0.001**	0.76	0.55, 0.99	0.074
Endocardial maxOWD	1.01	1.00, 1.02	**0.027**	1.01	1.00, 1.02	0.3	1.00	0.99, 1.02	0.4
Midmyocardial LV LS	0.78	0.70, 0.86	**<0.001**	0.78	0.70, 0.86	**<0.001**	0.78	0.70, 0.86	**<0.001**
Midmyocardial maxOWD	1.01	1.00, 1.01	0.083	1.01	1.00, 1.03	0.2	1.00	0.99, 1.01	0.6
Epicardial LV LS	0.76	0.67, 0.85	**<0.001**	0.60	0.44, 0.77	**<0.001**	0.74	0.48, 1.03	0.11
Epicardial maxOWD	1.01	1.00, 1.01	**0.030**	1.01	1.00, 1.02	0.2	1.00	0.99, 1.01	0.8
LVEF	0.96	0.93, 0.99	**0.008**	0.95	0.86, 1.04	0.3	1.06	0.99, 1.16	0.11
Z-score LVEDd	1.18	1.05, 1.34	**0.008**	0.69	0.41, 1.13	0.2	1.14	0.97, 1.34	0.10
LVM/BSA	1.02	1.01, 1.03	**<0.001**	1.03	1.00, 1.06	0.092	1.01	1.00, 1.03	0.07

Absolute values of strain indices were used for statistical analysis. Cardiac arrhythmias included ventricular tachycardia, ventricular fibrillation, and atrioventricular blockage III°. *p*-value of <0.05 was considered statistically significant. CI—confidence interval; BSA—body surface area; LS—longitudinal strain; LV—left ventricular; LVEDd—LV end-diastolic diameter; LVEF—LV ejection fraction; LVM—LV mass; maxOWD—maximum opposing wall delay, OR—odds ratio.

## Data Availability

Data are available upon reasonable request to the corresponding author; the request must include a description of the research proposal.

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
