# Peer review of "Prognostic Value of Speckle Tracking Echocardiography-Derived Strain in Unmasking Risk for Arrhythmias in Children with Myocarditis"

_biomedicines, 2024, doi:10.3390/biomedicines12102369_

Round 1

Reviewer 1 Report

Comments and Suggestions for Authors

This retrospective study evaluates left ventricular (LV) myocardial deformation using

speckle-tracking echocardiography-derived LV longitudinal strain in pediatric patients

diagnosed with myocarditis, using data from the "MYKEE" registry. The methodology is

comprehensively described, with blinded analysis performed by senior cardiologists. The

study includes 175 children and examines the prognostic value of impaired LV longitudinal

strain, finding a strong correlation with reduced left ventricular ejection fraction (LVEF) and

arrhythmias.

1. The study design is retrospective, with only about one-quarter of the registry

population eligible for this analysis. A notable strength is the confirmation of

myocarditis diagnosis via MRI and/or endomyocardial biopsy, adhering to standard

protocols. The inclusion and exclusion criteria are clearly defined and visually

represented in the flowchart. The authors have thoroughly described the methodology,

including blinded analysis by senior cardiologists, and the results are effectively

illustrated. The discussion is well-constructed, and the conclusions are supported by

the data. While not novel, the findings are of interest to this patient population.

2. Introduction: The introduction is well-crafted, providing essential background and

justifying the need for this analysis.

3. Study Population: The criteria for selecting the study population are adequately

detailed. The confirmation of myocarditis diagnosis using MRI and/or

endomyocardial biopsy ensures adherence to standard protocols. The inclusion and

exclusion criteria are clearly outlined and depicted in the flowchart.

4. Data Collection, Echocardiography, and Statistical Analysis: The methodology is

well-delineated, with blinded analysis conducted by senior cardiologists. Additionally,

the authors have addressed inter- and intra-observer agreement using the Bland-

Altman method.

5. Results: The results are effectively illustrated.

6. Discussion and Conclusion: The discussion is well-developed, and the conclusions are

appropriately supported by the data.

Major Comments

ï‚· Only 175 out of 772 patients were included in the study. Can the authors provide

baseline characteristics of the excluded patients, and do they resemble those of the

included cohort?

ï‚· Non-sustained ventricular tachycardia (VT) is not typically considered a malignant

arrhythmia. Could the authors clarify their definition of malignant arrhythmia, provide

a reference, or modify the terminology to specify sustained VT, ventricular fibrillation

(VF), or third-degree atrioventricular (AV) block?

ï‚· How do the Kaplan-Meier curves appear after excluding non-sustained VT?

ï‚· The reviewer acknowledges the documentation of arrhythmias during hospitalization,

but how were arrhythmias tracked after discharge? Please clarify and expand on the

methodology.

ï‚· How were events, particularly cardiac deaths, adjudicated? Was this done through

independent adjudication?

ï‚· Although the study primarily focuses on LV systolic function, the manuscript does not

mention diastolic dysfunction. Examining the relationship between LV diastolic

dysfunction and strain could provide additional insights.

ï‚· It is noteworthy that LV longitudinal strain predicts arrhythmias even in patients with

preserved LVEF. However, cardiac MRI findings indicate higher late gadolinium

enhancement (LGE), suggesting more scar tissue in patients with preserved LVEF

compared to those with mildly reduced or reduced LVEF, with a statistically

significant difference. How do the authors account for this observation?

Reviewer 2 Report

Comments and Suggestions for Authors

Authors presented a well-based methodology study regarding the evaluation of LV myocardial deformation using speckle-tracking echocardiography (STE)-derived longitudinal strain (LS) and its diagnostic and prognostic value in children with myocarditis. The introduction, methods, results and discussion are well presented. The only point that is missing is data regarding the follow-up of these patients.

Comments on the Quality of English Language

Minor grammar editing is required.

Reviewer 3 Report

Comments and Suggestions for Authors

Peer Review Report

Title: Prognostic value of speckle tracking echocardiography-derived strain in unmasking risk for arrhythmias in children with myocarditis

The submitted manuscript is a well-written, insightful, and important contribution to the field of pediatric cardiology, specifically focusing on myocarditis in children and the prognostic value of speckle-tracking echocardiography (STE). Below are my comments regarding the quality of the manuscript:

Strengths:

  1. Clear Research Question and Aim: The authors have clearly articulated the aim of the study, which is to evaluate the prognostic value of STE-derived longitudinal strain (LS) in predicting arrhythmias and major adverse cardiac events (MACE) in pediatric myocarditis. The rationale is well-grounded and addresses an important gap in the clinical management of pediatric patients with myocarditis.

  2. Methodology: The study design, including the use of a multicenter registry and large sample size (175 patients across 13 centers), strengthens the findings and adds to the robustness of the results. The statistical analyses used, including logistic regression and ROC analysis, are appropriate for the study objectives.

  3. Clinical Relevance: The manuscript highlights the significant role of layer-specific LS in predicting adverse outcomes in children with preserved left ventricular ejection fraction (LVEF), which is highly relevant for clinicians managing myocarditis in pediatric populations. This has the potential to influence clinical decision-making and improve early risk stratification.

  4. Data Presentation: The data are presented in a clear and comprehensive manner, with detailed tables summarizing the clinical characteristics of the cohort, echocardiographic findings, and statistical results. The figures effectively illustrate key points, such as the correlation between LS and LVEF, and the Kaplan-Meier curves add depth to the prognostic evaluation.

Areas for Improvement:

  1. Literature Context: While the introduction provides sufficient background, the discussion section could benefit from a more detailed comparison with other studies that have evaluated STE-derived strain in pediatric populations. This would help situate the findings within the broader body of research and highlight the novel aspects of this study.

  2. Limitations: The manuscript briefly mentions some limitations, but a more detailed discussion of the retrospective nature of the echocardiographic analyses, and potential variability in imaging quality across centers, would be beneficial. Furthermore, acknowledging the challenges in follow-up data, which could influence the prognostic value of LS, would provide a more balanced perspective.

  3. Future Directions: The study could be strengthened by including a brief section outlining potential areas for future research, such as the validation of strain imaging in other pediatric cohorts or exploring the utility of STE-derived strain in real-time clinical settings.

Conclusion:

This manuscript makes a valuable contribution to the field by demonstrating the prognostic utility of STE-derived LS in predicting adverse outcomes in pediatric myocarditis, particularly in patients with preserved LVEF. The findings have important clinical implications and are likely to be of significant interest to the readership of Biomedicines. I recommend the manuscript for publication with minor revisions as outlined above

Comments on the Quality of English Language

please edit some typos / mistakes

Reviewer 4 Report

Comments and Suggestions for Authors

In this retrospective multicentric study, the Authors demonstrated the incremental prognostic value of the STE-derived LS over LVEF for predicting both MACE and malignant arrhythmias in 175 pediatric patients with cardiac MRI- or EMB-confirmed myocarditis. 

Even in patients with preserved LVEF, an impaired LS was able to identify, among the pediatric patients affected by myocarditis, those at increased risk of malignant arrhythmias.

The manuscript is well written and each section is exhaustively presented.

Notably, the statistics is excellent and the results clearly exposed.

I congratulate the Authors with their work.

I have only one suggestion for the Authors:

A number of limitations of speckle tracking echocardiography (STE) that limit its use in the clinical practice should be aknowledged. The principal limitations of strain echocardiographic imaging are the intervendor variability, the dependence on on frame rate and the possible influence on intra- and inter-rater reproducibility of STE parameters exerted by extrinsic mechanical factors. The Authors could cite these references: PMID: 28528162 and PMID: 38231080
